# Ultrahigh and Tunable Electromagnetic Interference Shielding Performance of PVDF Composite Induced by Nano-Micro Cellular Structure

**DOI:** 10.3390/polym14020234

**Published:** 2022-01-07

**Authors:** Yang Yang, Shuiping Zeng, Xiping Li, Zhonglue Hu, Jiajia Zheng

**Affiliations:** Key Laboratory of Urban Rail Transit Intelligent Operation and Maintenance Technology & Equipment of Zhejiang Province, Zhejiang Normal University, Jinhua 321004, China; yanghqu@163.com (Y.Y.); bww2019@yeah.net (S.Z.); zhonglue.hu@zjnu.edu.cn (Z.H.); jiajia.zheng@zjnu.cn (J.Z.)

**Keywords:** composite, supercritical CO_2_ foam, cell size, EMI shielding

## Abstract

Lightweight and efficient electromagnetic interference (EMI) shielding materials play a vital role in protecting high-precision electronic devices and human health. Porous PVDF/CNTs/urchin-like Ni composites with different cell sizes from nanoscale to microscale were fabricated through one-step supercritical carbon dioxide (CO_2_) foaming. The electrical conductivity and electromagnetic interference (EMI) shielding performance of the composites with different cell sizes were examined in detail. The results indicated that the nanoscale cell structure diminishes the EMI shielding performance of the composite, whereas the microscale cell structure with an appropriate size is beneficial for improving the EMI shielding performance. A maximum EMI shielding effectiveness (SE) of 43.4 dB was achieved by the composite foams which is about twice that of the solid composite. Furthermore, as the supercritical CO_2_ foaming process reduces the density of the composite by 25–50%, the EMI SSE (specific shielding effectiveness)/t(thickness) of the composite reaches 402 dB/(g/cm^2^), which is the highest value of polymer foam obtained to the best of the authors’ knowledge. Finally, compression tests were performed to show that the composites still maintained excellent mechanical properties after the supercritical CO_2_ foaming process.

## 1. Introduction

In recent years, with the rapid development of 5G wireless communication technology, the spatial layout of communication equipment tends to be miniaturized, dense, and large-scale [1,2,3]. However, the resultant electromagnetic wave pollution can cause high-precision electronic devices to malfunction and even lead to human health problems. Therefore, lightweight, flexible, and efficient electromagnetic interference (EMI) shielding materials have always been hot research issues for scientists [4,5,6].

Typically, common metallic EMI shielding materials have drawbacks of high density, easy corrosion, and difficult shaping, which restricts its application in special fields such as aerospace and open-air environments [7,8,9]. Thus, new conductive polymer composites have been widely recognized by researchers for their unique features such as light-weight, corrosion resistance, and ease of molding [10,11,12]. At present, carbon-based conductive composite materials belong to the widely studied EMI shielding materials. Due to the low density, high aspect ratio, and excellent electrical conductivity, the carbon nanofillers such as carbon nanotubes, graphite sheets, graphene nanoplates, and carbon fibers, etc., allow composites with extremely low content to exhibit excellent EMI shielding performance. In addition, magnetic nano-fillers are used for embedding in conductive composites to improve their EMI shielding properties due to their magnetic loss ability in attenuating electromagnetic waves. For example, Zhao et al. [13] prepared PVDF (polyvinylidene fluoride)/CNTs (carbon nanotubes)/Ni flexible composite films by solution mixing and compression molding, whose maximum EMI SE (shielding effectiveness) reached 57.3 dB at K-Band, with a thickness of 0.6 mm. Lee et al. [14] fabricated PP (polypropylene)/NCCF (nickel-coated carbon fiber) composite material by mechanical mixing and injection molding, and its EMI SE was 48.4 dB at a frequency of 10 GHz. Some researchers have also improved the electromagnetic wave absorption capacity of composites by regulating the surface morphology of the magnetic particles [15,16].

The supercritical fluid foaming process can greatly reduce the density of the composites and fabricate a porous structure in the polymer matrix. Therefore, this technique is highly attractive as it possesses a number of benefits such as energy saving [17,18], environmental protection [19,20], thermal insulation [21,22], and vibration and noise reduction [23]. A large number of studies have reported that the introduction of the microcellular structure is conducive in improving the EMI shielding properties of conductive polymer composites [24,25,26]. Chul B. Park’s team [27] prepared an HDPE (high density polyethylene)/graphene composite foam through a supercritical fluid (N_2_) injection foaming process, which exhibited higher EMI shielding effectiveness compared to solid composites and reduced the density of the composites by 26%. Wang et al. [28] introduced a microcellular structure in PLA/graphite nanocomposites by an injection-molded foaming process, which improved the electrical conductivity and mechanical properties of the composite material, and its EMI shielding effectiveness reached 45 dB. Zhang et al. [29] fabricated lightweight PVDF/Ni chain composite foams with a uniform closed-cell structure using two-step batch-foaming. The as-prepared composite foams exhibited enhanced electrical conductivity (0.01 S/m) and EMI shielding effectiveness (26.8 dB). However, there are also a few studies reporting that EMI shielding performance was reduced after supercritical fluid treatment. Yang et al. [30] reported microcellular silicone rubber/MWCNTs (multi-walled carbon nanotubes)/Fe_3_O_4_ nanocomposites by supercritical carbon dioxide treatment. The batch-foaming process improved the microwave-absorbing ability but diminished the EMI shielding effectiveness of the sample from 45 dB to 27.5 dB. Zhang et al. [31] fabricated lightweight and multifunctional PMMA (polymethyl methacrylate)/Fe_3_O_4_@MWCNTs composite foams using the supercritical carbon dioxide foaming process. The EMI shielding effectiveness of PMMA/10 wt%Fe_3_O_4_@MWCNTs composite decreased from 25.11 dB to 16.0 dB after the supercritical carbon dioxide foaming process. The disparity in the EMI shielding performance after supercritical fluid treatment indicated that the differences in cell structure in the composite foams may cause the EMI shielding properties of the composite to differ. However, most studies focused on the effects of different nano-fillers on the foaming and EMI shielding properties of composites, while little is known in detail on the effects of cell structure with different sizes on the EMI shielding properties of composites.

Herein, to obtain composites with excellent electrical conductivity and EMI shielding performance and the mechanical properties of the PVDF/10 wt%CNTs/10 wt% Ni composites, the effect of cell structure on the composites was studied in detail. The urchin-like Ni powder was synthesized by a hydrothermal method. Then, the PVDF/10 wt%CNTs/10 wt% Ni composites prepared by solution mixing and injection molding were placed in an autoclave with different saturation temperatures for the supercritical carbon dioxide foaming process. By changing the saturation temperature, the cell structure from nanoscale size to micro scale size of the composite foam is adjustable. Next, the effects of different cell structures in the composite foams on the electrical conductivity and EMI shielding properties were investigated. It is worth noting that the PVDF/10 wt%CNTs/10 wt% Ni composites treated with supercritical carbon dioxide exhibited both enhanced electrical conductivity and EMI shielding performances. More importantly, the EMI SSE (specific shielding effectiveness) of the composite achieves 402 dB/(g/cm^2^), which is by far the highest ever reported.

## 2. Materials and Methods

### 2.1. Raw Materials

Nickel sulfate hexahydrate (NiSO_4_∙6H_2_O), sodium hydroxide, 85% hydrazine hydrate (N_2_H_4_∙H_2_O) and N,N-dimethylformamide (DMF) were purchased from Sinopharm Chemical Reagent Co., Ltd. (Shanghai, China). All reagents were of analytical grade and used without further purification. Multi-walled carbon nanotubes (MWCNTs, NC7000) were purchased from Nanocyl (Sambreville, Belgium), with an average diameter of 9 nm and a length of about 1.5 μm. The polyvinylidene fluoride 6020 (MRF: 2 g/10 min, 230 °C/21.6 kg) was provided by Solvay (Brussels, Belgium) and the CO_2_ gas (purity: 99.5%) supplied by DaTong gas Co., Ltd. (Chengdu, China).

### 2.2. Synthesis of Urchin-like Nickel

According to the preparation process of An et al. [32], their study improved the synthesizing method of urchin-like nickel. Typically, first 5 mmol of nickel sulfate hexahydrate was dissolved in a flask containing 50 mL of deionized water. Then, 10 mL of hydrazine hydrate and 0.12 g of NaOH were added. Subsequently, the mixture was magnetically stirred for 10 min to obtain a uniform mixture. After that, the prepared precursor was transferred into a 100 mL Teflon reactor and kept at 100 °C for 4 h. Finally, the reactant was washed several times with deionized water and ethanol and separated with a permanent magnet to obtain urchin-like nickel.

### 2.3. The Preparation of PVDF/CNTs/Urchin-like Ni Composites

Flexible PVDF/CNTs/urchin-like Ni composites were prepared by solution casting and injection molding, as shown in Figure 1. First, PVDF particles were mixed with N,N-dimethylformamide solution at a ratio of 1 g/10 mL. After magnetically stirring at 60 °C for 3 h, a PVDF homogeneous solution was obtained. Then 10 wt% of MWCNTs was added to the above mixed solution for mechanical stirring for 30 min. Next, urchin-like Ni powders were added and ultrasonically shaken for 30 min to obtain a uniform PVDF/CNTs/urchin-like Ni mixture. Subsequently, the PVDF mixed solution was transferred to an evaporation dish and placed in a vacuum oven to dry at 60 °C for 5 h. Finally, the dried PVDF/CNTs/urchin-like Ni composite films were cut off and placed in an injection molding machine to prepare a rectangular parallelepiped sample. The injection process parameters are shown in Table 1.

### 2.4. PVDF/CNTs/Ni Composites Batch-Foaming Process

The microcellular PVDF/CNTs/Ni composites were prepared using a homemade batch foaming device, as shown in Figure 1. The batch foaming device consists of a high-pressure syringe pump (ISCO 260D, Lincoln, NE, USA), an Omron temperature controller, an autoclave equipped with a temperature sensor and a ring heater, and a pressure-relief ball valve. In a typical one-step foaming process, the cavity of the autoclave was quickly purged with CO_2_ gas 3 times to remove the air in the autoclave. Then the sample with a size of 14 mm × 6 mm × 4 mm was put into the autoclave. Next, the gas pressure in the storage tank of the high-pressure syringe pump was kept at around 3000 psi. When the temperature inside the autoclave achieved the desired value through the temperature controller, the autoclave was filled with supercritical CO_2_ for 30 min to ensure the composite absorb CO_2_ completely. After the adsorption finished, the pressure-relief ball valve was quickly opened to release the high-pressure CO_2_ in the autoclave. Finally, the autoclave was immersed in a cold-water bath for rapid cooling to obtain a PVDF/CNTs/Ni composite foam.

To investigate the effect of different cell structure on the EMI properties of microcellular PVDF/CNTs/Ni composites, the composites were foamed in the saturation temperature range of 164 °C to 172 °C, with an interval temperature of 1 °C. The foamed samples were named F-164 to F-172 in this order.

### 2.5. Characterization

The crystal structure of the solid composite was examined by an X-ray diffractometer (D8 Advance, Bruker AXS, Karlsruhe, Germany) using a Cu target (λ = 0.15418 nm). The density of the foamed sample was measured by a buoyancy method using a high-precision electronic balance (Sartorius, Goettingen, Germany) with a density measuring accessory. The expansion ratio (*φ*) of the sample was calculated from the density of the solid sample (*ρ_s_*) and the density of the foamed sample (*ρ_f_*) in accordance with Formula (1):(1)$$\varphi =\frac{\rho_{s}}{\rho_{f}},$$

The microscopic morphology and elemental information of the as-received urchin-like Ni and the cross-sectional morphology of the composite foams were observed using a field emission scanning electron microscope (FE-SEM, S-4800, HITACHI, Tokyo, Japan) equipped with an energy spectrometer (Oxford Instruments, Oxford, UK). Before observation, the cross sections of the composite foams were obtained by immediately fracturing the composites after 10 min of immersing in liquid nitrogen, and platinum was sprayed on the surface of the cross section to eliminate the charging effect. After receiving the SEM images of the cross-section, the average cell size was measured by an open-source image processing software called *ImagePy* [33]. The number of counted cells is not less than 100. The cell density (N_0_) of the composite was calculated by the following formula:(2)$$N_{0}={\left(\frac{n}{A}\right)}^{3/2}\times \varphi ,$$
where n is the number of cells in an area of A (cm^2^) in the SEM image.

The electrical conductivity is defined as $\sigma =L/\left(S\times R\right)$ (where *L* is the length of the sample, *S* is the cross-sectional area of the sample, and *R* is the volume resistance of the sample). A high-precision Digital multimeter DMM 4040 (Tektronix, Beaverton, OR, USA) carried out the measurement of the resistance value (*R*) of the composite foam. Each sample was measured 5 times to reduce the measurement error. The compression properties of the composite foams with different expansion ratios were tested on a universal material testing machine (UTM4202, SUNS, Shenzhen, China) according to ASTM-D1621. The compression specimen was a cubic specimen with a side length of 5 mm, which was cut from the fabricated PVDF/CNTs/Ni foam. The compression speed of the crosshead during the measurement was 1 mm/min.

The EMI shielding performance of the as-prepared composite foams at different saturation temperatures was measured using a network vector analyzer (VNA, Agilent N5324A, Beijing, China) equipped with a waveguide in the K-band (18.0–26.5 GHz) frequency range. Each sample was a standard cuboid specimen with a size of 10.6 mm × 4.3 mm × 2 mm (length × width × thickness) made by cutting and grinding. The main evaluation parameters of the EMI shielding performance of the composites were as follows: reflectivity (*R*), transmissivity (*T*), *SE*_Total_ (*SE_T_*), *SE*_Reflection_ (*SE_R_*) and *SE*_Absorption_ (*SE_A_*), calculated from the forward reflection coefficient (*S*_11_) and reverse transmission coefficient (*S*_12_) measured by VNA according to the following formulas [29]:(3)$$R={\left|S_{11}\right|}^{2}={\left|S_{22}\right|}^{2},$$
(4)$$T={\left|S_{21}\right|}^{2}={\left|S_{12}\right|}^{2},$$
(5)$$SE_{R}=-10\log_{10}\left(1-R\right),$$
(6)$$SE_{A}=-10\log_{10}\left(\frac{T}{1-R}\right),$$
(7)$$SE_{T}\approx SE_{R}+SE_{A},$$

## 3. Results and Discussion

### 3.1. X-ray Diffraction Patterns and Element Analysis

The morphology and elemental information of urchin-like Ni synthesized by the liquid phase reduction method were analyzed by SEM and EDS technology, as shown in Figure 2. The surface morphology of urchin-like Ni is shown in Figure 2a, in which the inset is an SEM image of urchin-like Ni at higher magnification. It can be seen from the figure that the overall diameter of urchin-like Ni is relatively uniform at low magnifications, with a diameter of about 1 to 2 μm, and the length of acicular Ni on the surface of the Ni particles is about 200 nm at higher magnifications. Figure 2b is the point-scanning energy spectrum of the urchin-like Ni in Figure 2a. Overall, Ni is the major element in the sample, with few impurity elements C, O, Al. From the relative atomic content table in Figure 2b, the atomic ratio of Al and O is approximately 2:3, which is likely the Al_2_O_3_ compound. Therefore, the impurity element could come from the aluminum-based conductive carbon tape which is used to fix the Ni powder sample. In conclusion, it can be confirmed that there was no oxidation in the prepared urchin-like Ni sample.

Figure 3 exhibits the XRD patterns of neat PVDF and PVDF/CNTs/Ni solid composite foam. First, the X-ray diffraction peaks of neat PVDF mainly appear at 2θ = 17.6°, 18.4°, 20.0°, and 26.7°, which correspond to the (100), (020), (110), and (021) crystal planes of the PVDF α-phase, respectively [34,35,36]. After adding nanofillers, there was no change in the position of the diffraction peaks of PVDF, but the intensity of the diffraction peaks of PVDF crystal decreased obviously. The XRD result indicated that the addition of nanofillers did not transform the PVDF α-phase into β-phase but blocked the formation of PVDF α-phase. In addition, since the position of the diffraction peak of CNTs is at 2θ = 26.5°, which is very close to the diffraction peak of the PVDF α-phase (021) crystal plane, the diffraction peak of CNTs was therefore not shown in the pattern. Compared with the XRD pattern of neat PVDF, the XRD diffraction peaks of PVDF/CNTs/Ni composite show three extra diffraction peaks at 2θ = 44.5°, 51.8°, and 76.4°. According to the JCPDS no. 04-0850, these peaks correspond to the (111), (200), and (220) crystal planes of urchin-like Ni, respectively [37]. Moreover, there are no other strong diffraction peaks other than these three diffraction peaks, which indicated that urchin-like Ni was not oxidized during the composite processing.

### 3.2. Cell Morphology of the PVDF/CNTs/Urchin-like Ni

Figure 4 displays a cross-sectional morphology of PVDF/10 wt% CNTs/10 wt% Ni composites foamed at different saturation temperatures (164–172 °C). As the saturation temperature increases, the cell morphology of the composite foams gradually become uniform. Because PVDF is a semi-crystalline polymer, the local melt strength of the crystalline and amorphous regions of the polymer would be distinct at different saturation temperatures [38]. In the low temperature range (164–165 °C), it is difficult for the cells in the polymer matrix to grow after nucleation, because the overall melt strength of the composites is too high. Eventually, nano-cells formed in the composite. When the saturation temperature increased to 166 °C, microscale cells begin to form in some amorphous regions as the melt strength of the composite decreased. When the saturation temperature further increased from 167 °C to 169 °C, the cell morphology of the composite foams became more uniform and the cell diameter gradually increased. This is due to the crystalline region of the composites beginning to melt at the higher temperature. However, when the saturation temperature was between 170 °C and 172 °C, the cell size of the composite foams decreased with increasing temperature. This is because the melt strength is too low, causing the cells to collapse and fuse.

The statistical results of the expansion ratio, density, cell size, and cell density of the PVDF/10 wt% CNTs/10 wt% Ni composites foamed at different saturation temperatures are shown in Figure 5. As can be seen from Figure 5a, in the foaming temperature range (164–172 °C), the minimum density of the composite foams is 0.515 g/cm^3^, which is reduced to 27.2% of the original density. Compared with neat PVDF foaming performance [28], the addition of urchin-like Ni and CNTs reduces the maximum expansion ratio of the composite foams, which is due to the addition of urchin-like Ni increasing the viscosity of the composites and further impairing cell elastic expansion capacity. It can be seen from Figure 5b that as the saturation temperature gradually increases, the cell diameter tends to grow slowly, and the cell density decreases accordingly. The maximum cell diameter and the minimum cell density were 7.26 μm and ${1.24\;\times \;10}^{10}{\;\mathrm{cells}/\mathrm{cm}}^{3}$, respectively.

### 3.3. Electrical and EMI Shielding Properties of the PVDF/CNTs/Ni Foams

Figure 6 shows the electrical conductivity and EMI shielding performance of PVDF/10 wt% CNTs/10 wt% Ni composites foamed at different saturation temperatures. Figure 6a illustrates the bar graph of the electrical conductivity of the composite foams as a function of the saturation temperature. Overall, the electrical conductivity of the composite foams initially decreases, then increases, and finally decreases which can be explained from the CNT distribution of the representative SEM images, as shown in Figure 7. In comparison to the solid PVDF/CNTs/Ni composite with an electrical conductivity of 0.06 S/cm, the composite foamed at a low temperature shows a decreased electrical conductivity due to the isolation effect of nano-scale cells on the conductive fillers. With the saturation temperature increasing from 164 °C to 166 °C, the electrical conductivity of the composite foams picks up slightly, which results from the enlarged cells in the composite foams. In the above process, one notable turning point is that the electrical conductivity of the composite foaming at 166 °C exceeds that of the solid composite. The reason is that the microcellular cells begin to appear inside the foamed composite at this temperature, and this was proved by the SEM images shown in Figure 7. Furthermore, when the saturation temperature further increased from 166 °C to 169 °C, the electrical conductivity of the composite foams significantly increased, with a maximum of 0.214 S/cm, which is about 250% higher than that of the solid composite. This is due to the expanding micro scale cells causing the isolated conductive fillers to concentrate on the cell walls, as is also shown in Figure 7. Meanwhile, it also indicates that micro scale size cells are more effective than nano scale size cells in constructing enhanced conductive networks. However, when the saturation temperature further increases, the electrical conductivity of the composite foams starts to decrease. Such a decrease originates from the loss in density of the conductive filler in the cell wall caused by the decreasing cell sizes. In summary, only proper cell sizes could improve the conductive properties of the composites.

Generally, the EMI shielding effectiveness is used to characterize the composites’ ability to inhibit microwave radiation, which can be calculated through $\mathrm{SE}=\text{–}10\log_{10}(P_{\mathrm{in}}{/P}_{\mathrm{out}})$. Usually, SE = 20 dB shows that the material is capable of blocking 99% of microwave radiation, which is now the minimum design level of commercial EMI shielding material. The EMI shielding effectiveness of PVDF/10 wt% CNTs/10 wt% Ni composites foamed at different saturation temperatures in the K-band (18–26.5 GHz) is shown in Figure 6b. As can be observed from the figure, the EMI shielding effectiveness of each sample does not change remarkably with increasing microwave frequency, which reveals that the composite foams exhibit independent frequency properties. In order to intuitively examine the changes in the EMI shielding effectiveness of the foamed composites at different saturation temperatures, each part of the EMI shielding effectiveness, i.e., *SE_T_*, *SE_A_*, and *SE_R_* of all samples at 22 GHz were separately extracted and compared, as shown in Figure 6c. Overall, the trend of the EMI shielding effectiveness of the composite foams with increased temperature is roughly the same as its electrical conductivity. The sudden decrease in the EMI shielding effectiveness of the F-167 sample was likely due to an accidental error during the test, which, however, does not affect the overall trend. With the saturation temperature increasing gradually, the EMI SE first increased from 13.4 dB to 43.4 dB, then decreased to 22.4 dB, and finally rose to 24.4 dB. Compared with the solid sample, the EMI shielding effectiveness of the foamed sample improved significantly, up to 82%. Notably, the absorption shielding loss (*SE_A_*) of all samples is much larger than their reflection shielding loss (*SE_R_*). For instance, the *SE_T_*, *SE*_A_, and *SE_R_* values of the sample F-169 are 43.4 dB, 38.4 dB, and 5.0 dB, respectively. The absorption loss accounts for 88.5% of the total loss, which indicates that the composites treated by supercritical fluid still behave as the dominant EMI shielding mechanism. In addition, the reflection loss values (*SE_R_*) of the composite foams are basically maintained at about 4.5 dB, and the change trend of the absorption loss value (*SE*_A_) is consistent with the total shielding effectiveness trend. Thus, it can be concluded that the effect of the cell structure on the EMI shielding performance of the composite foams is mainly reflected in the change of its absorption loss, which can be explained by two aspects. On the one hand, the cell structure reconstructs the conductive path by changing the distribution of the conductive nanofillers in the polymer matrix, which has a vital influence on the internal current caused by microwave radiation. Meanwhile, the variation of the internal current changes the microwave losing ability of the composites [39,40]. On the other hand, the presence of the cell structure leads to the formation of an impedance mismatch interface between CO_2_ gas and composite inside the composite, which results in multiple reflection loss during the transmission of the microwave [41,42].

Considering the strict restrictions on material density and thickness in practical applications, the EMI specific shielding effectiveness divided by the thickness (EMI SSE/t) was used to characterize the relative EMI shielding effectiveness of the PVDF/CNTs/Ni composites. Thus, the EMI SSE/thickness values of the composites foamed at different saturation temperatures were calculated and results are shown in Figure 6d. On the whole, the EMI SSE/t value increases first and then decreases as the saturation temperature increases, with a maximum value of 402.0 dB/(g∙cm^2^) at 169 °C. Moreover, the EMI SSE/t values of all foamed samples are improved in comparison with that of the solid composite, due to the reduced density of the composites. The max EMI SSE/t value of the sample is five times that of solid composite. The result indicates that the EMI shielding property and density of the composite foam could be adjusted according to practical application by controlling the cell morphology of the composite foam.

In order to visually describe the microwave attenuating loss process when the microwaves incident the composites, a schematic diagram of the microwave attenuation mechanism of foamed and solid composite is displayed in Figure 8. When the electromagnetic wave irradiates the composite, reflection loss would occur at the interface between the external space and the composite. This is due to the impedance mismatch between air and the composite [43,44,45]. After microwave transmission in the composite, there are three main losses. First, the conduction loss that occurs when carriers excited by electromagnetic wave radiation conduct and transition [46,47]. Second, due to the high frequency alternating electric field of electromagnetic waves, interfacial polarization loss is caused by rapid exchange of excited charges between the conductive fillers (Ni and CNTs) of the composite [48,49]. Third, when magnetic nanoparticles are contained in the composite, the alternating magnetic field of the electromagnetic waves causes magnetic domain resonance and natural eddy currents of the magnetic substance to generate magnetic loss [50,51]. Obviously, after the composite material is processed by the supercritical CO_2_ foaming process, the microwave attenuation loss changes in regard of two aspects. On the one hand, the proper cell structure can improve the conductive path of the composite material, thereby improving the electrical conductivity of the composite which could increase the conductive loss of the composite [52]. On the other hand, because there is an impedance mismatch interface between the CO_2_ gas and the cell wall, multiple interface reflection losses take place on the cell wall, which improves the ability of the composite foam to absorb electromagnetic waves [53,54].

To further highlight the lightweight and high-efficiency EMI shielding performance of the prepared PVDF/CNTs/urchin-like Ni composite foams, the EMI shielding effectiveness and EMI SSE/t values of various porous composites in recent years are listed in Table 2. It can be seen that the porous composite prepared in the current experiment exhibits a superior EMI shielding performance compared to other porous materials, with a thickness of only 2 mm. In particular, the composite foam exhibits an efficient EMI shielding property at low density, which results in an ultrahigh EMI SSE/t value of 402.0 dB/(g∙cm^2^), higher than other materials reported in the literature. Therefore, the design and preparation method for the composite foam in this work achieved the adjustment of the EMI shielding property of the composite by changing the cell structure, which has direct significance for the performance regulation of large injection foam products. Meanwhile, it also can be applied in the fields of architecture, microelectronic devices, and aerospace.

### 3.4. Compression Properties of the PVDF/CNTs/Ni Foams

Figure 9a demonstrates the compressive stress–strain curves of PVDF/CNTs/Ni composites foamed at different saturation temperatures. It can be seen that the curves are not completely separated, which is caused by the non-uniform distribution of the cell structure in the sample. The rigid foam stress–strain curve is usually divided into three stages: linear elastic stage, yield platform, and compactness [61]. Because the cell diameter of PVDF/CNTs/Ni composite is too small, the elastic deformation phase of the cell wall is very short, which causes the compression stress–strain curves of the composite foams only to show the compaction compression phase. Therefore, the actual factor affecting the compressive strength of PVDF/CNTs/Ni composites is the solid content of the composites per unit volume. To clearly observe the compressive strength of the composite foams with different cell morphologies, the compressive strength at strain of 30% was taken out for comparison, and the bar graph is shown in Figure 9b. As can be seen, with the foaming temperature gradually increasing, its compressive strength first generally decreases and then increases slightly. The composite foamed at 164 °C exhibits the highest compressive strength of 19.4 MPa. As the saturation temperature increases, the cell size gradually expands from the nanometer level to the micron level. Meanwhile, the compressive strength of the composite material gradually decreases as the solid content of the composite foam per unit volume gradually decreases. Then, as the temperature saturation further increases, the expansion ratio of the composite foam gradually decreases, and the cell walls become thicker. As a result, the solid content of the composite material per unit volume begins to increase again and thus the compressive strength of the composite foam picks up slightly.

## 4. Conclusions

In this work, as-synthesized urchin-like Ni and CNTs were distributed in a PVDF matrix through a casting method and injection molding to obtain a high-efficient EMI shielding composite. Then, supercritical carbon dioxide foaming treatment was performed at different saturation temperatures to fabricate porous PVDF/CNTs/Ni composites with different cell sizes. The effect of cell structure on the electrical conductivity, the EMI shielding properties of composites, and the mechanical properties were studied. It was found that the nano scale cell structure causes the EMI shielding effectiveness of the composite foam to be lower than that of the solid composite, and the micro scale cell structure with an appropriate size helps to improve the EMI shielding effectiveness of the composite. Moreover, although the introduction of urchin-like Ni and CNTs reduced the foaming properties of the composite, it improved the electrical conductivity and magnetic property of the composite to a certain extent, which resulted in an improved EMI shielding effectiveness of the composite foams to 43.4 dB and a maximum EMI SSE/t of 402 dB/(g/cm^2^). Finally, the mechanical properties of composite foams were investigated to determine their application prospects in engineering. The compression test shows that the composite material can maintain a compressive strength of 3.5 MPa at a high expansion ratio. Therefore, the supercritical CO_2_ foaming process can improve the EMI shielding effectiveness of the composites while maintaining excellent compression performance under the appropriate process parameters.

## Figures and Tables

**Figure 1 polymers-14-00234-f001:**
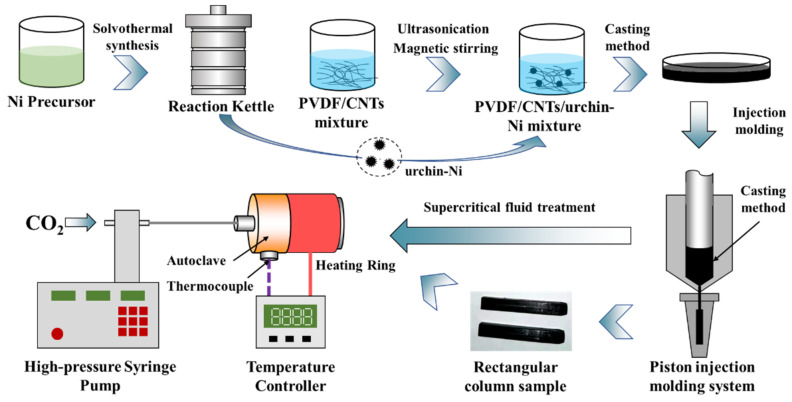
The preparation processing of PVDF/10 wt% CNTs/10 wt% urchin-like Ni compo-site foams.

**Figure 2 polymers-14-00234-f002:**
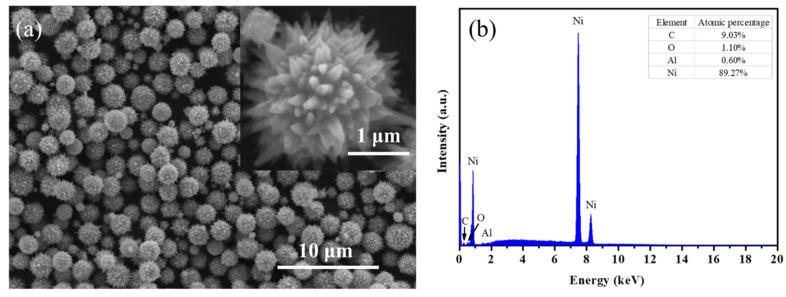
The SEM image (**a**) and EDS pattern (**b**) of the as-prepared urchin-like Ni particles.

**Figure 3 polymers-14-00234-f003:**
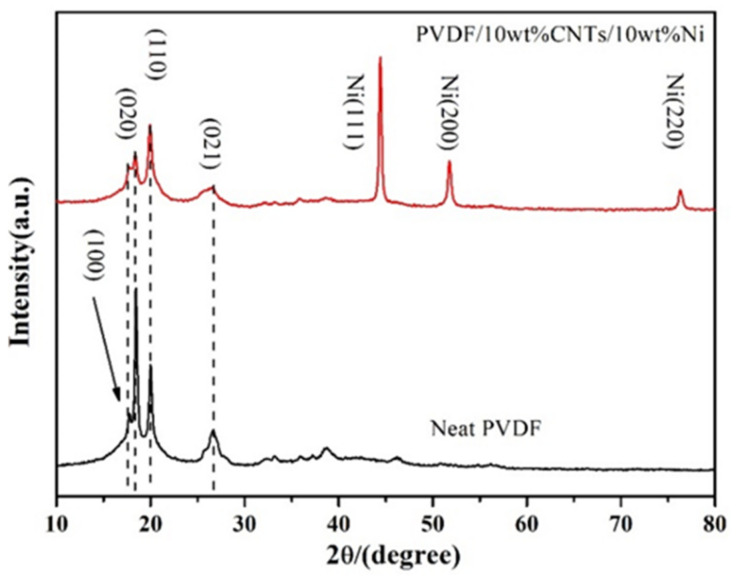
The XRD patterns of neat PVDF and PVDF/CNTs/urchin-like Ni composite.

**Figure 4 polymers-14-00234-f004:**
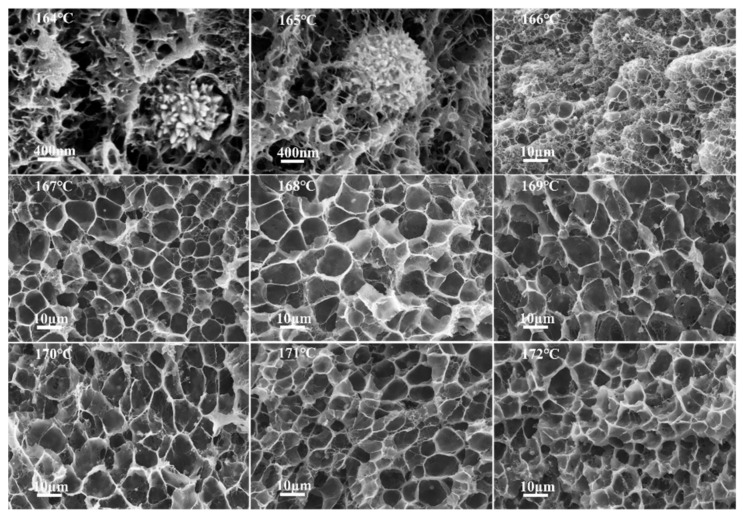
The SEM images of the cross-section of the PVDF/10 wt% CNTs/10 wt% Ni composites foaming under different temperature (164–172 °C). The pictures of 164 °C and 165 °C were taken at a magnification of 20,000, and the others were taken at a magnification of 2000.

**Figure 5 polymers-14-00234-f005:**
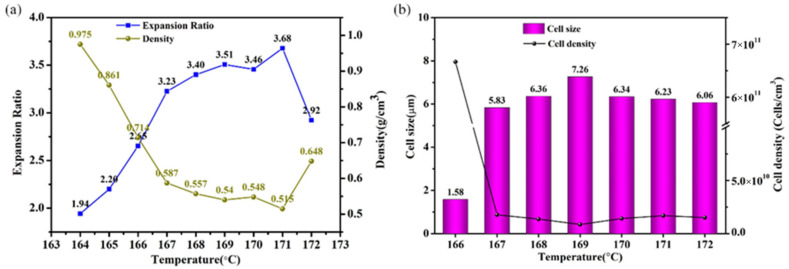
(**a**) The density and expansion ratio of the PVDF/10 wt% CNTs/10 wt% Ni foamed at different temperatures; (**b**) The cell size and cell density of the PVDF/10 wt% CNTs/10 wt% Ni foamed at different temperatures.

**Figure 6 polymers-14-00234-f006:**
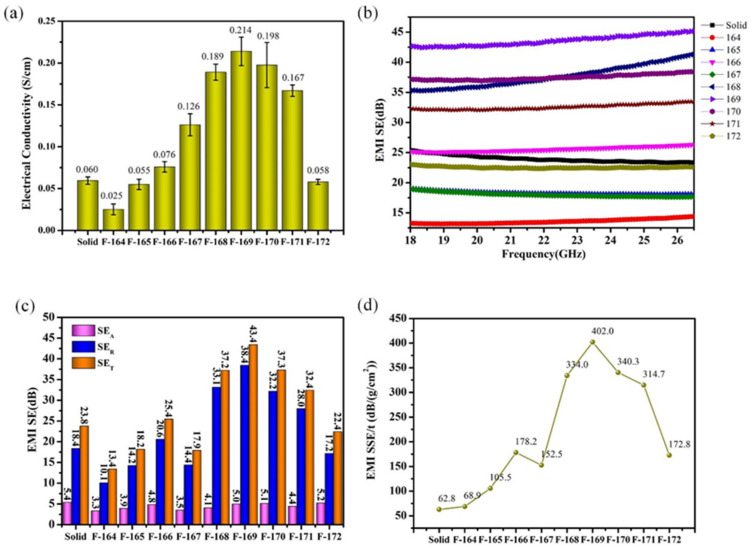
(**a**) The electrical conductivity of PVDF/10 wt% CNTs/10 wt% Ni composites foamed at different temperature; (**b**) The EMI shielding effectiveness of PVDF/10 wt% CNTs/10 wt% Ni foams at the frequency range of (18.0–26.5 GHz); (**c**) The *SE_T_*, *SE_A_*, *SE_R_* of PVDF/10 wt% CNTs/10 wt% Ni foams at the frequency of 22 GHz; (**d**) The EMI SSE/thickness of PVDF/10 wt% CNTs/10 wt% Ni composites foamed at different temperature.

**Figure 7 polymers-14-00234-f007:**
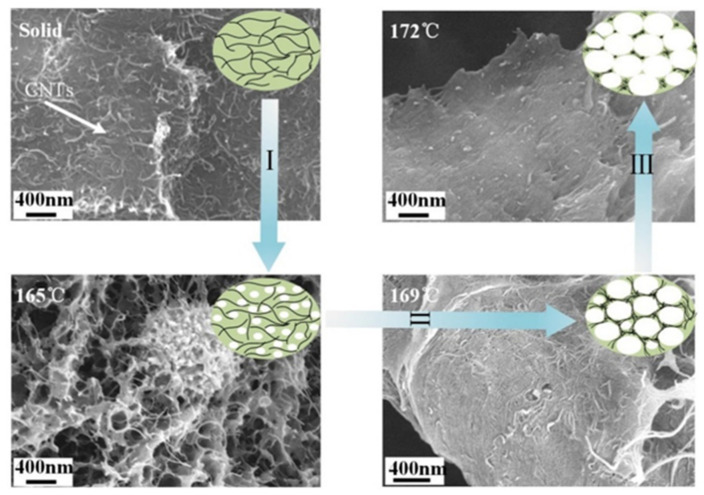
The SEM images of CNT distribution in solid, F-165, F-169, and F-172 samples; The inset in each image is a schematic diagram of the CNT distribution.

**Figure 8 polymers-14-00234-f008:**
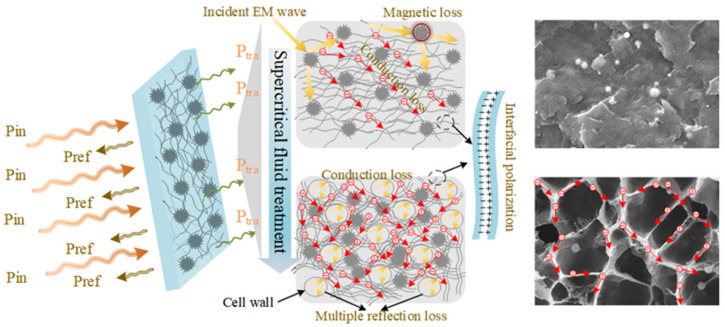
The schematic diagram of the microwave shielding mechanism in the solid and foamed PVDF/CNTs/urchin-like composites.

**Figure 9 polymers-14-00234-f009:**
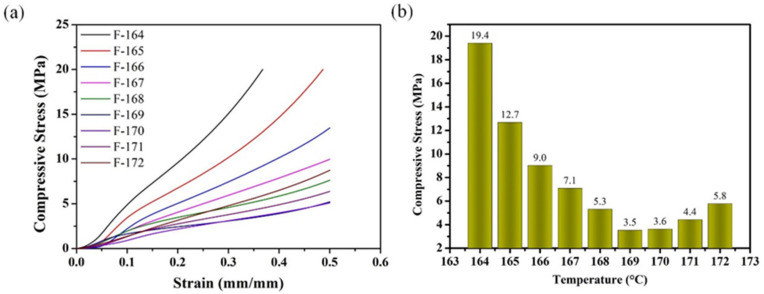
(**a**) The compressive stress-strain curves of PVDF/CNTs/Ni composites foamed at different saturation temperatures; (**b**) The compressive strength of composite foams at 30% strain.

**Table 1 polymers-14-00234-t001:** Injection process parameter settings.

Parameter Item	Set Value
Cylinder temperature (°C)	245
Mold temperature (°C)	80
Injection pressure (MPa)	70
Cooling time (s)	10
Packing pressure (MPa)	60
Packing time (s)	10

**Table 2 polymers-14-00234-t002:** The EMI shielding effectiveness of porous composites reported in recent years.

Matrix	Filler Loading	Thickness (mm)	EMI SE (dB)	EMI SSE dB/(g/cm^3^)	EMI SSE/t dB/(g/cm^2^)	Ref.
PVDF	15 wt% MWCNT	2.0	56.72	71.79	358.95	[25]
PMMA	2 wt% MWCNT + 1 wt% GnP	2.0	15.7	27.07	135.35	[54]
PLLA	10 wt% MWCNT	2.5	23	77.00	308.00	[55]
Nature rubber	6.4 wt% MWCNT	1.3	33.74	40.65	312.69	[56]
PVDF	10 wt% GnPs	2.5	27	29.67	118.68	[57]
PVDF	16 wt% MWCNT	2.0	28.5	45.97	229.85	[9]
TPU	6.5 wt% RGO	1.8	21.8	18.16	100.89	[58]
PP	1.1 vol% stainless-steel fiber	3.1	48	75.0	241.94	[59]
PEI	10 wt% graphene@Fe_3_O_4_	2.5	16.6	41.5	166.0	[60]
PVDF	10 wt% CNTs + 10 wt% Ni	2.0	43.4	80.4	402.0	This work

## Data Availability

The datasets generated and analyzed during the current study are available from the corresponding author upon reasonable request.

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
