# Peer review of "Ultrahigh and Tunable Electromagnetic Interference Shielding Performance of PVDF Composite Induced by Nano-Micro Cellular Structure"

_polymers, 2022, doi:10.3390/polym14020234_

Round 1

Reviewer 1 Report

The manuscript entitled " Ultrahigh and tunable electromagnetic interference of PVDF composite induced by nano-micro cellular structure" includes a lot of work and contains many data that are originally synthesized and interpreted. The works fits in your journal rigors and I recommend its publication.

I made the following comments:

  • In the Introduction, the objectives of the study are not sufficiently stated;
  • Many abbreviations(PVDF, CNT etc.) are not explained

Reviewer 2 Report

The research work carried out by the authors is interesting and hopefully will contribute to the scientific community. The manuscript is well written and explained with relevant data, though there are some sentences weakly constructed.  My comments are listed below sequentially for author's kind information.

Line 40: Please revise the word abroad by widely or whatever the authors like.

Line 47: Please illustrate PVDF (polyvinylidene fluoride) at its first appearance in the text and also mention MW in materials section.

Line 104: Please revise sentence with better English

Line 108: “uniform mixture” will look better instead of “uniform mixed solution.”  

Figure 1: Is “rectangular column sample” product? There is no arrow sign to correlate samples with the diagram.

Line 118: As the authors have taken only 10 wt.% CNTS so I think they can mention exact amount instead of the words “certain amounts”.

Overall comments on the manuscript:

  • Authors have embedded 10 wt.% of CNT and 10 wt.% Ni in PVDF matrix and examined mainly the effect of saturation temperature (164-172 °C) on the EMI shielding efficiency. I didn’t find any information showing the reason of choosing the narrow temperature range and the mentioned amount of CNTs and Ni particle. . Moreover, the effect of variable amount of CNTs and Ni particles on EMI shielding would be additional information.
  • Authors are requested to cite references in supporting formulas (3-7) used to calculate EMI SEs. Also need to mention clearly what the letters mean.
